


# Still normal? Contextualizing real-time data with long-term statistics to monitor anomalies and systematic changes in storm surge activity – Introduction of a prototype web tool storm surge monitor for the German coasts

Xin Liu, Insa Meinke, and Ralf Weisse

Institute of Coastal System Analysis and Modeling, Helmholtz-Zentrum Hereon, Max-Planck-Str. 1, 21502 Geesthacht, Germany

*Correspondence to*: Ralf Weisse (ralf.weisse@hereon.de)

**Abstract.** Storm surges represent a major threat to many low-lying coastal areas in the world. While

most places can cope with or are more or less adapted to present-day risks, future risks may

increase from factors such as sea level rise, subsidence, or changes in storm activity. This may

require further or alternative adaptation and strategies. For most places, both forecasts and real-

time observations are available. However, analyses of long-term changes or recent severe extremes

that are important for decision-making are usually only available sporadically or with substantial

delay. In this paper, we propose to contextualize real-time data with long-term statistics to make

such information publicly available in near real-time. We implement and demonstrate the concept of

a "storm surge monitor" for tide gauges along the German North Sea and Baltic Sea coasts. It

provides automated near real-time assessments of the course and severity of the ongoing storm

surge season and its single events. The assessment is provided in terms of storm surge height,

frequency, duration, and intensity. It is proposed that such near real-time assessments provide

added value to the public and decision-making. It is further suggested that the concept is

transferable to other coastal regions threatened by storm surges.

## 1 Introduction

High water levels at the coast are typically caused by the combination and interaction of different

factors. These include, for example, the astronomical tide, the effects of strong winds pushing the

water towards the coast (wind or storm surge), or seasonal, interannual, and long-term mean sea

level changes. Depending on the region, nonlinear interaction among the different factors occurs

and may substantially contribute to the extremes and enhance the risks (Arns et al., 2017). For

example, in shallow water, the efficiency of the wind in producing the surge may vary substantially

with tidal water levels (phase of the tide), and the propagation of the tidal wave may in turn depend

on surge levels (Horsburgh and Wilson, 2007).



Extreme sea levels resulting from such processes pose a major risk to many low-lying coastal areas worldwide that are at least seasonally affected by storms. The most deadly and devastating storm
surges were caused by tropical cyclones. Examples comprise the storm surge generated by the 1970 Bhola Cyclone in Bangladesh that caused approximately 300,000 casualties or the 2005 Hurricane Katrina storm surge which represents one of the most costly natural disasters in US history (Needham et al., 2015). Extratropical storm surges, although less severe, still bear a substantial threat.

In the mid-latitudes, parts of the North Sea and Baltic Sea coasts are examples of regions highly susceptible to extreme sea levels. For instance, in 1953 and 1962 two major disasters occurred at the North Sea coast. Both flooded several thousand hectares of land and killed several hundred or thousands of people (Gönnert and Buß, 2009; Hall, 2013). In 1872, the Danish and German Baltic Sea
coasts were devastated by an extreme storm surge, which still represents the highest on record (Feuchter et al., 2013).

Since then, coastal defense was improved significantly. Particularly in the North Sea, higher storm surges than those reported in 1953 and 1962 were observed in more recent years. For example, the
storm surge in 1976 caused higher water levels than in 1962 at many gauges at the German North Sea coast. In the recent past, the extratropical storm Xaver in 2013 caused exceptionally high water levels along the entire coastline of Lower Saxony (Deutschländer et al., 2013; Rucińska, 2019), including the cities of Hamburg and Bremen. Contrary to the devastating events of 1953 and 1962, later extremes caused no severe damages or casualties due to significantly improved coastal
protection. Due to the latter, public perception of vulnerability and risk has decreased in recent years (Ratter and Kruse, 2010). Nevertheless, the risk still exists, and it may further increase in the expected course of anthropogenic climate change (Gaslikova et al., 2013; Wahl, 2017; Weisse et al., 2014).

For the German North Sea coast, there is a considerable number of studies analyzing either variability and/or long-term changes in extreme sea levels (e.g. Dangendorf et al., 2014). Such studies focus on either the description of past and present (e.g. Weisse and Plüß, 2006) or possible future (e.g. Gaslikova et al., 2013) variability and change in general or link them to some driving mechanisms (e.g. Woodworth et al., 2007). Studies based on observations (Dangendorf et al., 2014)
as well as on taking modeling approaches (e.g. Vousdoukas et al., 2016; Woth et al., 2006) or


statistics (Butler et al., 2007) do exist. The main conclusion from the majority of these studies is that extreme sea levels along the German North Sea coast have increased over the past about 100 years. Primarily, this is suggested to be a consequence of rising mean sea level. Changes in the wind climate produced some interannual and decadal variability but no noticeable trend (e.g. Weisse et

al., 2012). Changes in the tidal regime may also have contributed to some extent (e.g. Hollebrandse, 2005). For the Baltic Sea, several studies carried out similar analyses. Based on gauge records, most authors conclude that observed increases in extreme Baltic Sea levels are mainly due to corresponding increases in the mean sea level of about 10–15 cm within the last century (Marcos and Woodworth, 2018; Meinke, 1999; Ribeiro et al., 2014; Weisse and Meinke, 2016). Some of the

northernmost gauges, however, deviate from that picture due to decreasing relative sea level caused by Glacial Isostatic Adjustment (GIA) of the Earth's crust (Barbosa, 2008; Ribeiro et al., 2014). Strong correlations of Baltic Sea level variability with the large-scale atmospheric circulation are reported as well (Hünicke and Zorita, 2006; Karabil, 2017; Karabil et al., 2017). For the future, present studies indicate a further increase in extreme sea levels mainly in response to rising mean

sea level, while storm-related contributions exhibit considerable uncertainty with the upper bound suggesting an increase of up to a few decimeters (e.g. Vousdoukas et al., 2017; Weisse et al., 2012). At some places, the figure may be substantially modified by vertical land motions such as subsidence or uplift (e.g. Ribeiro et al., 2014; Richter et al., 2012).

For decision-making, information on long-term changes in storm surge activity is of utmost importance (e.g. Kodeih, 2019; Kodeih, et al., 2018; Weisse et al., 2015). Mostly, detailed local information is requested, and the contextualization of the ongoing storm surge activity is increasingly asked for (e.g. Meinke, 2017; Weisse et al., 2020). This includes, for example, questions on physically plausible upper limits, on probabilities for co-occurrences of storm surges and other

hazards (e.g. river floods in estuaries) or, often in the immediate aftermath of an event, on the extent to which this event was "normal" or can be attributed to anthropogenic influences such as climate change. The latter requires contextualizing events within a detection and attribution framework; that is, to first provide information on how usual or unusual an event has been (detection) and second to attribute causes to unusable cases (attribution) (e.g. Hegerl, 2010).

Unfortunately, the update of such information occurs more or less sporadically or at specific time intervals in the order of years. From our experiences in collaborating with decision-makers and other stakeholders, we suggest that near real-time availability of such localized information and its contextualization may provide substantial added value to professional stakeholders or the public discussion in general.


In the following, we present an approach on how such information may be contextualized and can

be made publicly available in near real-time. The approach is initially developed for storm surges

along the German North and Baltic Sea coasts and initially focuses on detection only. We propose

that the approach can be easily transferred to other regions and extended to other variables and/or

attribution. In Sect. 2, we introduce the approach, which in the following will be referred to as Storm

Surge Monitor. This section also describes the data, methods, and functionalities. In Sect. 3, the

long-term development of storm surge activity at the German coasts is analyzed. Moreover, it

exemplarily provides a retrospective analysis of the storm surge seasons 2018/19 and 2019/20.

Discussion and summary are provided in Sect. 4.

## 2 The Storm Surge Monitor

### 2.1 General Concept

Information on sea level extremes is typically available and used in several different ways. For

example, real-time data are used to monitor and forecast extremes; that is, to be prepared and to

protect inhabitants, assets and infrastructures of coastal regions. Potential long-term changes, which

need to be assessed to adopt risk management procedures or coastal protection measures, are

typically not assessed in real-time, but become available only at fixed intervals (typically several

years) or in the aftermath of an exceptionally extreme case when specific analysis has been carried

out. Assessment of extremes regarding the extent to which they were unusual and if they may serve

as examples of ongoing long-term changes is therefore only possible with considerable delay. How

can these two different sources of information be linked together in near real-time? To address this

issue, we propose a concept in which real-time measurements are putting into context with

observed long-term conditions; that is, their statistics such as mean conditions, variability, expected

extremes, or long-term changes. By minimizing delays between the occurrence of extremes and

their climatological assessment, we propose to add value to monitoring and assessing ongoing storm

surge seasons and their single events.

The concept was implemented and tested in a prototype web tool, which is referred to as Storm

Surge Monitor (or shortly as the Monitor) in the following. It was developed for several frequently

considered measures such as the height, frequency, duration, or intensity of extreme events. The

long-term, seasonal and single event assessments are provided. The Monitor is available in both

English (www.stormsurge-monitor.eu) and German (www.sturmflut-monitor.de).


### 2.2 Tide gauge data

We used tide gauge data from the German North Sea and Baltic Sea coasts to set up the Monitor.

Based on the availability of historical and real-time data, in total 10 tide gauges along the German
North and Baltic Sea coasts were selected (Fig. 1).

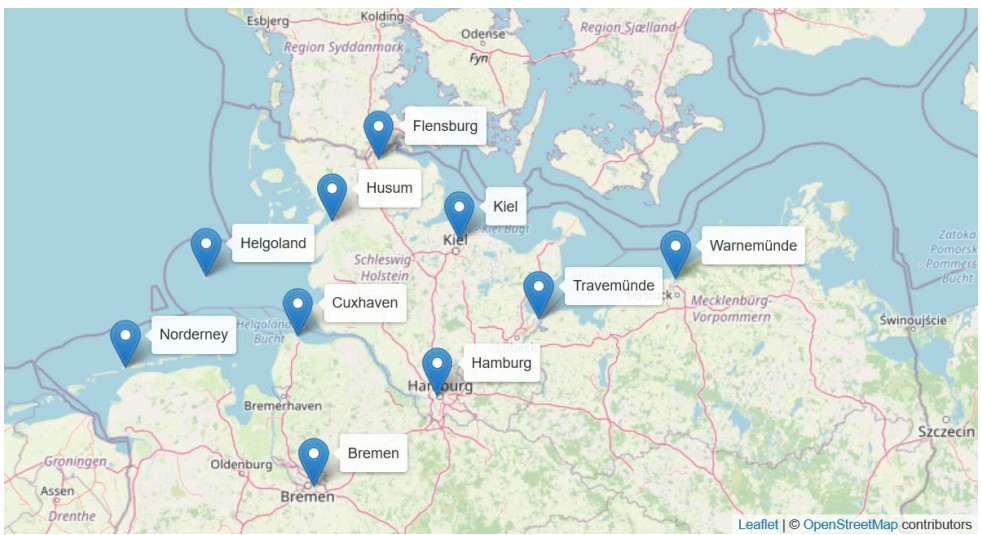

**Figure 1.** Location of the tide gauges. (Credit: Leaflet | Maps © OpenStreetMap contributors.

Distributed under a Creative Commons BY-SA License)

The data processing workflow is illustrated in Fig. 2. Two types of data, historical and real-time, were
used. The historical tide gauge records were used to derive long-term statistics. The available record
length, temporal resolution, and source of the data vary (Table 1). Depending on the data availability

and the record length, we used twice-daily high water level or hourly data. While the advantage of
using hourly data is the ability to provide information on the duration and intensity of extreme
events, the maxima of extremes may be underestimated due to the sampling frequency. Only
records starting not later than the 1950s were selected.


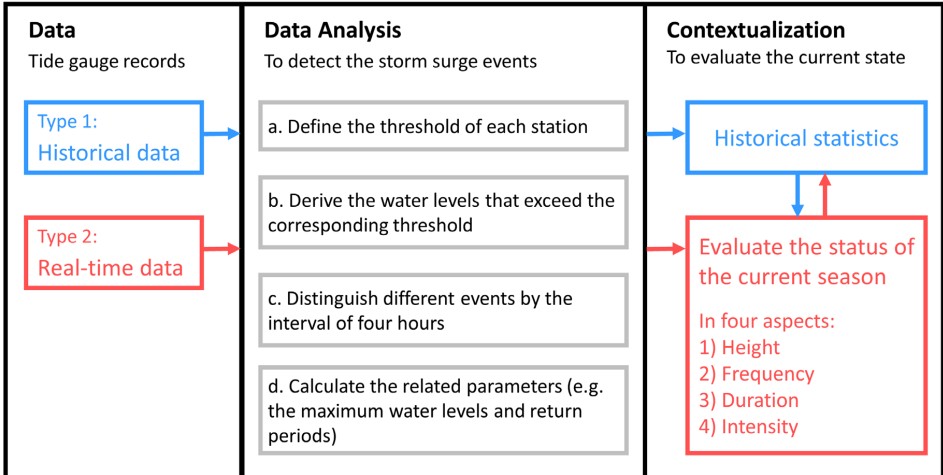

**Figure 2.** Schematic diagram of the data processing workflow.

**Table 1.** Summary of the tide gauge records used to calculate the long-term statistics.

| Tide gauge | Period | Temporal resolution | Data source |
|---|---|---|---|
| **Husum** | 1936–2018 | High water | German Federal Waterways and Shipping Administration (WSV), provided by the German Federal Institute of Hydrology (BfG) |
| **Helgoland Binnenhafen** | 1953–2018 | High water | WSV, provided by BfG |
| **Cuxhaven** | 1901–2018 | High water | WSV, provided by BfG |
| | 1919–2018 | Hourly | University of Hawaii Sea Level Center (UHSLC) (*Caldwell et al.*, 2015) |
| **Hamburg St. Pauli** | 1951–2019 | High water | WSV, provided by BfG |
| **Bremen Weserwehr UW** | 1954–2019 | High water | WSV, provided by BfG |
| **Norderney** | 1901–2018 | High water | WSV, provided by BfG and the Coast Research Center (FSK) of the Lower Saxony State Agency for Water, Coastal and Nature Conservation (NLWKN) |
| **Flensburg** | 1955–2019 | Hourly | WSV, provided by BfG |
| **Kiel-Holtenau** | 1955–2019 | Hourly | WSV, provided by BfG |

| Travemünde | 1950–2018 | Hourly | WSV, provided by BfG |
| Warnemünde | 1954–2018 | Hourly | WSV, provided by BfG |

To assess the characteristics of the ongoing storm surge season and the severity of the latest events,
the real-time data are needed. For this, the data available every minute from PEGELONLINE
(www.pegelonline.wsv.de) were used. They were automatically fetched four times daily and
resampled to high water levels or hourly values depending on the availability of historical data at
each tide gauge.

The observed water level from tide gauges is measured relative to a land-based reference (relative
sea level) and contains contributions from atmospheric and oceanic dynamics as well as from solid
Earth processes (Stammer et al., 2013). For coastal management, it is the relative sea level that is
important (Rovere et al., 2016; Stammer et al., 2013). In our analyses, the data are therefore
intentionally not decomposed or detrended to retain contributions from sea level rise or subsidence.

**2.3 Near real-time information delivery**

To ensure fast information delivery, the Monitor is automatically updated four times a day (at 1:30,
7:30, 13:30, and 19:30 CET). When the water level at any gauge exceeds a given threshold, a new
event at this particular gauge is recognized. Accordingly, new plots are automatically generated to
provide the latest information about the event and its long-term and seasonal contextualization.
While this near real-time availability of an assessment represents the purpose of the Monitor, it has
some limitations on the other hand. Since the real-time data are normally raw data from
measurements, it is possible and unavoidable that, for example, due to instrument failures during a
storm no or only errournes data are accessible, which could potentially affect the detection and
classification of an extreme. Also, other technical problems could occur, leading to unusable values
over extended periods that can last for hours or months. For example, the tide gauge Norderney
experienced a technical problem from May to September 2018. The measurements during that
period are invalid with random high/low values or no value at all, and the notification of the
incorrect measurements was posted on the PEGELONLINE website. To deal with such erroneous data
in real time, a quick quality control algorithm was implemented that marks and removes values
whose minute increments were higher than 0.3 m (spikes) and values that were not within a
reasonable range (1–12 m). When updated quality controlled data become available, figures may
thus change compared to the near real-time assessment.



### 2.4 Detection of storm surge events

To identify and assess the severity of storm surges, the first step is to define the threshold. There are

various methods defining the threshold of storm surge in practice and in research, e.g. from the DIN

4049 (DIN 4049-3, 1994) or from the Federal Maritime and Hydrographic Agency of Germany

(Bundesamt für Seeschifffahrt und Hydrographie, BSH) (Gönnert, 2003). The definition from the DIN

4049 is based on the statistical probability of occurrence, while the one from BSH is relatively simple

to understand and more related to the public, as the BSH is responsible to issue a public warning of

storm surges. Therefore, the definition from the BSH was used in this study. For the German North

Sea coast, a storm surge event is detected when the water level exceeds the mean tidal high water

level (MThw) by at least 1.5 m. The event is further classified as a severe or very severe event when

its maximum water level exceeds the MThw by at least 2.5 m or 3.5 m. Because of the different

atmospheric and oceanographic conditions, storm surges at the German Baltic Sea coast are divided

into four classes. Events of which the maximum water level exceeds the mean water level (MW) by

1.00–1.25 m are referred to as storm surge events. Events with water levels of 1.25–1.50 m above

MW or 1.50–2.00 m above MW are referred to as medium or severe events. Cases in which the

maximum water level is higher than 2.00 m above MW are referred to as very severe events. In the

Monitor, the MThw at the North Sea and the MW at the Baltic Sea tide gauges were calculated over

the standard reference period 1961–1990 defined by the World Meteorological Organization (WMO)

so that all events (past and present) at each gauge refer to the same reference (Table 2). Also, the

period 1961–1990 is referred to as the reference period in the analyses.

**Table 2.** List of the mean tidal high water levels (MThw) and the mean water levels (MW) (in m

relative to NHN) of the reference period 1961–1990 for the tide gauges along the North Sea and

Baltic Sea coasts.

| North Sea | | Baltic Sea | |
|---|---|---|---|
| Tide gauges | MThw | Gauges | MW |
| **Husum** | 1.58 m | **Flensburg** | -0.02 m |
| **Helgoland Binnenhafen** | 1.07 m | **Kiel-Holtenau** | 0 m |
| **Cuxhaven** | 1.46 m | **Travemünde** | 0.02 m |
| **Hamburg St. Pauli** | 1.93 m | **Warnemünde** | 0 m |
| **Bremen Weserwehr UW** | 2.47 m | | |
| **Norderney** | 1.14 m | | |


### 2.5 Definition of storm surge seasons and statistical analyses

At the German North Sea and Baltic Sea coasts, storm surge activity is most pronounced in the
winter season from October to March (Jensen and Müller-Navarra, 2008). In the Monitor, the period
from July to June of the following year was used to define a storm surge season, and all analyses and
statistics were computed based on these seasons. Each storm surge season is denoted by the year in
which the season ends to indicate the latest possible information. For instance, the period from July

2018 to June 2019 is referred to as season 2018/19 and marked as 2019 in the annual plots. The
course of a season and the monthly distributions are also derived and shown from July to June of the
following year. When the ongoing season ends, new long-term statistics are automatically
computed, in which the statistics of the terminated season will be included.

The steps of data processing are shown by the second column entitled Data Analysis in Fig. 2. The
first step is to derive the threshold of each gauge. Then, storm surge events are detected when
water levels exceed the corresponding threshold. It is possible to have multiple threshold
exceedances or pauses during one event. For this, we use a de-cluster interval of four hours to
separate one event from another. To estimate the probabilities of storm surges, we apply the

generalized extreme value (GEV) distribution to the annual maximum values (Hennemuth et al.,
2013). The parameters of the distribution (location, scale, and shape) are derived by using the
maximum-likelihood estimation (MLE). A wide range of the commonly used statistical distributions
and methods exists for extreme value studies, such as the Gumbel distribution together with block
(annual) maxima method, or the Generalized Pareto distribution (GPD) together with a peak over

threshold approach (e.g. Muis et al., 2016; Wahl et al., 2017). All these methods have their own
merits and can provide good estimations of extreme values. Thus, the selection among the methods
is often a trade-off and largely depends on the purpose of the study. As one type of the GEV, the
Gumbel distribution has only two parameters (location and scale) and is often used in global-scale
research, while a known limitation is that it could underestimate or overestimate the high end of

extreme values (Buchanan et al., 2017; Wahl et al., 2017). In this regard, the GEV and GPD are more
flexible shaped to better represent the extreme values. Both distributions are considered to have
stable and reliable performance for tide gauges along the German coast, while the GPD is suggested
when the dataset is short (Arns et al., 2013). Studies have compared various sampling methods in
detail, varying the block maxima method with r-largest sampling and the peak over threshold

method with different threshold selection (Arns et al., 2013; Wahl et al., 2017). The annual maxima
method is used in this analysis because it is robust to temporal and spatial variations as well as



suitable for the region. The annual maximum is defined as the maximum water level that occurs in an annual period, which is a storm surge season in this analysis.

## 2.6 Structure of the Monitor and main functionalities

The Monitor consists of ten tide gauges so far. Six of them are located at the German North Sea coast, and four of them are at the German Baltic Sea coast. The map (Fig. 1) of the Monitor is interactive, so users can access the information by clicking on the tide gauges. On the web page of a particular tide gauge, users can view figures, texts and interpretations that illustrate the long-term development of storm surge activity and the characteristics of the current season in comparison to

the historical seasons and long-term statistics. Depending on the real-time and historical data availability, different measures and statistics are accessible for each tide gauge via the navigating items. Information on height and frequency is presented for all gauges, while information on duration and intensity is provided only for the gauges with hourly data, i.e., Cuxhaven, Flensburg, Kiel, Travemünde, and Warnemünde. For each measure, two figures are displayed: the first one

shows the time development of that measure over the past few decades, and the second one allows an assessment on whether or not the current storm surge season can be considered unusual when compared to the long-term statistics of the past seasons. Underneath each figure, we provide a brief caption and an example interpretation to assist users in reading the figures. These elements complement each other to reveal a comprehensive description and contextualization of the ongoing

storm surge season and recent events.

To illustrate the main functionalities of the Monitor, the figures of season 2019/20 at the tide gauge Cuxhaven are discussed exemplarily (Figs. 3 to 5). Two important indicators addressing the height of extremes are assessed, namely the annual maximum water level and return period (Fig. 3).

Specifically, the maximum water level of the current season (red dot) is compared with the variations of the annual maxima of the previous seasons (black curve) (Fig. 3a). The plot allows for a quick visualization of the observed long-term changes and the extent to which the highest water level in an ongoing season is unusual. For easy visualization, the current season is highlighted in all the annual plots by the red-dotted line. The gray line provides an estimate of the linear trend with

the gray shaded area representing the 95 % confidence interval. The trend line is shown in solid when the estimated trend is significantly different from zero at the level of 95 % and dashed otherwise. The scale on the left axis is labeled as water level relative to the German reference surface (Normalhöhennull, NHN), while the right axis is labeled in reference to the severity

classification of storm surges (water level relative to MThw for the North Sea coast or relative to

MW for the Baltic Sea coast).

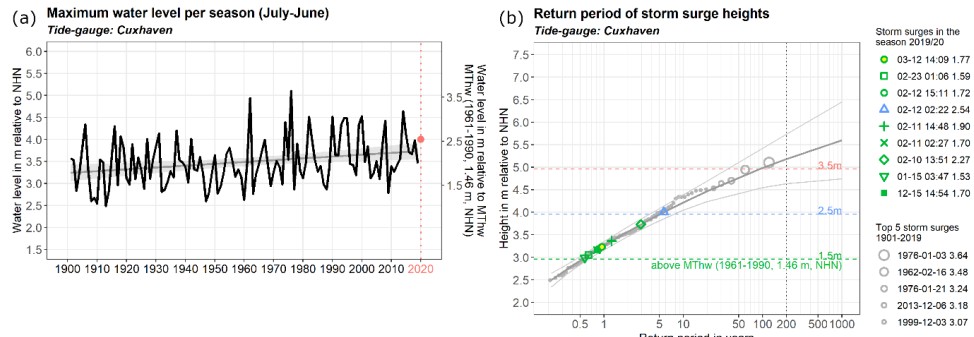

**Figure 3.** (a) Maximum water level per season and its linear trend at Cuxhaven. The trend (gray line)
is shown with the 95 % confidence interval (light gray band) and as a solid line when it is significant

at the level of 95 %. The red dot denotes the maximum water level observed in the current season
(season 2019/20 here). (b) Return period of storm surges at Cuxhaven. The events of the current
season are marked by colored symbols according to their heights and the respective severity
classifications (green – minor; blue – severe; red – very severe). The estimated distribution (dark
gray curve) of the annual maxima (gray points) over the previous seasons was derived using the

generalized extreme value (GEV) distribution and the maximum-likelihood estimation (MLE). The
gray band shows the 95 % confidence interval of the estimation. The top five severe historical events
in the available period are represented by gray open circles with size indicating their magnitudes.

The second plot provides an estimate of the return periods of the storm surges that has occurred in

the current season (Fig. 3b). Return periods are widely used to estimate the likelihood and severity
of extreme events (*Haigh et al.*, 2015; *Wahl et al.*, 2017). For example, if an event is referred to as a
one in a hundred years event, this means that such an event has a 1:100 (1 %) chance of happening
in any given year, no matter when the last similar event has happened. Figure 3b displays the
extreme value distribution (dark gray curve) estimated from historical data with its 95 % confidence

interval (two light gray lines) derived by fitting a GEV to the observed annual maxima. This fit is used
to evaluate the return period of the latest (ongoing) storm surges in near real-time. When an event
occurs, it will be placed on the top of the current season list and marked by a colored symbol whose
color denotes its severity (green – minor; blue – severe; red – very severe). In the example season,
two minor events with return periods less than five years can be identified (Fig. 3b). To further put

the recent events into perspective, a list of the five severest historical events during the available





period (seasons 1901–2019 here) is given underneath the list of the current season. These are represented by gray open circles with different sizes indicating their magnitudes.

Similarly, two plots assess the frequency of storm surges from a long-term perspective and within the current season (Fig. 4). For the long-term perspective, the number of storm surges per season is shown together with its trend (Fig. 4a). The severity of the events is marked with different colors to show the number of events in the different severity classes in each season. The current season is also highlighted in red as in Fig. 3a. While still ongoing, it could already and preliminarily be put into context with previous seasons, long-term variability, and change.


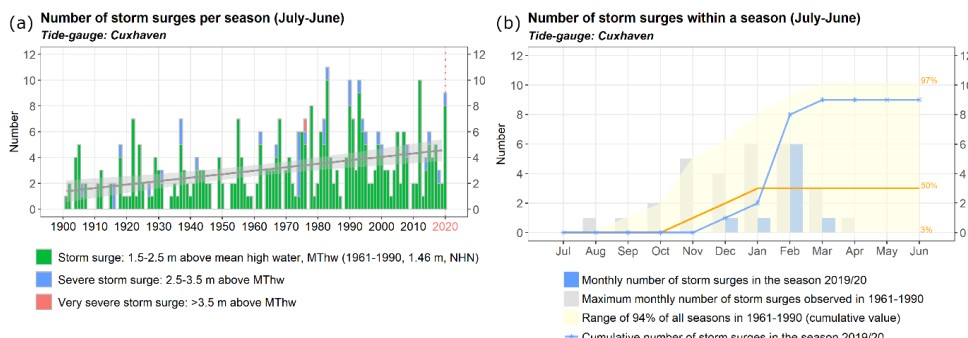

**Figure 4.** (a) Number of storm surges per season and its linear trend at Cuxhaven. The color of the bars denotes the degree of severity (green – minor; blue – severe; red – very severe). The trend (gray line) is shown with the 95 % confidence interval (light gray band) and as a solid line when it is
significant at the level of 95 %. The current season (season 2019/20 here) is highlighted with the red-dotted line. (b) Number of storm surges within a season (July–June) at Cuxhaven. Bars: the monthly number in the current season is shown as blue bars, and the historical monthly maximum in the reference period 1961–1990 is shown as gray bars. Curves: the cumulative number of events in the current season is shown as a blue curve, in comparison with the 50th percentile (orange curve) and
the range of the 3rd and 97th percentiles (yellow shaded area) over the reference period.

The second plot (Fig. 4b) was designed to illustrate the course of an ongoing season. For example, if the onset of a storm surge season is very early or late, if there is an unusual number of events within a particular month, or if the whole season is unusually active in terms of frequency and annual cycle.
The analysis is shown for a storm surge season from July to June of the following year. The number of events in each month of the current season (blue bars) is compared with the maximum number of events in the corresponding month over the reference period (1961–1990, gray bars). The blue curve





shows the cumulative number of events from the beginning of a particular current season until the

day of the website visit. The reference is given by the 50th percentile (orange curve) and the range

from the 3rd to the 97th percentiles (yellow shaded area) of the reference period. For example,

when the blue curve remains below the orange one, this indicates that fewer events than usual were

observed so far. The blue curve above the orange line but still within the yellow shaded area

suggests that the frequency of such a season is still within a normal range but belongs to more active

seasons. If the blue curve exceeds the yellow shaded area, indications for an exceptionally active

season exist.

In addition to height and frequency, duration and intensity are widely used measures to describe

characteristics of storm surges (Cid et al., 2016; Zhang et al., 2000) because height/frequency and

duration/intensity of events do not necessarily share a similar variability even though they are

related to some extent. Moreover, these two measures are also important from the perspective of

coastal protection and risk assessment (e.g. Kodeih et al., 2018). Therefore, information on duration

and intensity statistics are included in the Monitor, although they can only be provided for some tide

gauges where long-term hourly data are available. In the following analyses, duration denotes the

number of hours for which the water level exceeds the given storm surge threshold, while intensity

refers to the area between the water level measurements and the storm surge threshold.

Regarding storm surge duration, in particular the seasonal sum is a key indicator for coastal erosion.

On the other hand, looking at single events, the maximum intensity of storm surges in a season is

also critical for potential damages at coastal infrastructure. Thus, the total duration and the

maximum intensity of storm surges in each season are shown from a long-term perspective (Figs. 5a

and 5b). The current season is highlighted with an orange bar to be easily separable from the

previous seasons (gray bars). Assessment of the single events within a season is provided in Figs. 5c

and 5d, in which the current events (red dots) are shown relative to the monthly distributions

derived from historical data. The historical reference (blue) is illustrated in the form of a box plot

bounded by the 3rd and 97th percentiles. The median is given by the solid horizontal blue line, and

the historical maximum in each month is denoted by the blue dot. Assessment can be obtained from

the relative positions of the red dots, which indicate whether an event lasted longer or was more

intense in comparision to the monthly statistics of the reference period.

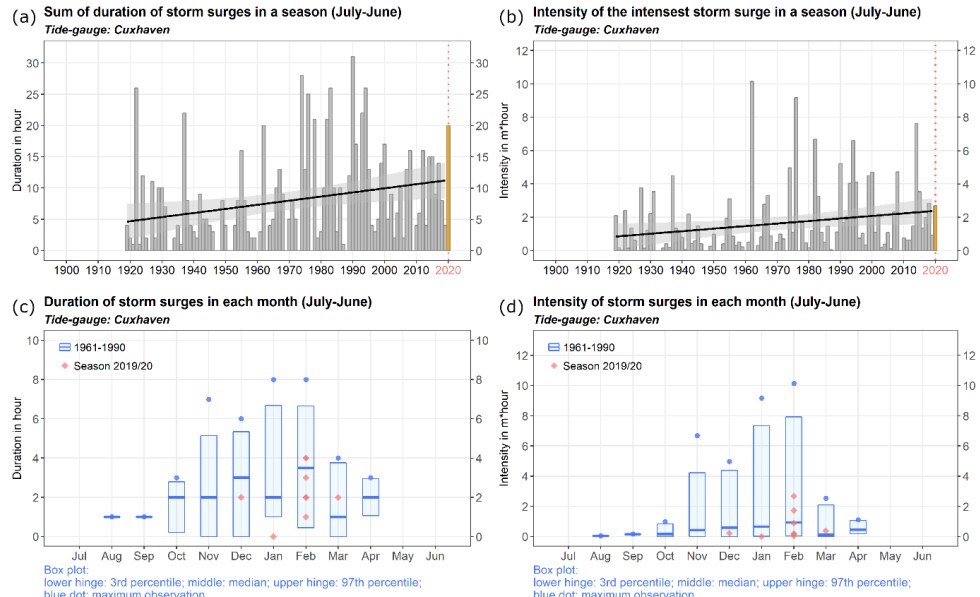


**Figure 5.** (a, b) Total duration and maximum intensity of storm surges per season at Cuxhaven. The gray bars represent the past seasons, while the orange bar denotes the current season (season 2019/20 here). The trend (black line) is shown with the 95 % confidence interval and as a solid line when it is significant at the level of 95 %. (c, d) Box plot of the duration and the intensity of storm

surges in each month at Cuxhaven. Blue shows the statistics (the 3rd, 50th, 97th percentiles, and the maximum) of the events in each month of the reference period 1961–1990, while the red dots signify the events of the current season.

## 3 Contextualizing real-time storm surge data with long-term statistics

In this section, we demonstrate the capabilities of the Monitor by analyzing the long-term

development over the available periods and the two recently concluded storm surge seasons, namely the seasons 2018/19 and 2019/20. Comparing the seasons, both in time and spatially as well as putting them into context with the historical distributions and changes, several inferences can be made. First, the anomalies and systematic changes in storm surge activity at each gauge can be inferred, which also reflects local differences among the tide gauges. Second, the consistency

between the observed anomalies and the long-term trend is evaluated in order to distinguish between randomly active seasons and active seasons that exemplarily fit into the long-term development. Third, differences between the North Sea and Baltic Sea coasts can be inferred, which reflect the different key processes and their settings contributing to the storm surges.


### 3.1 North Sea coast (incl. Elbe and Weser)

3.1.1 Long-term development of storm surges

*Height*

Annual maximum water level increased at all selected North Sea tide gauges over the available
periods (e.g. Figs. 3a and 6). Except for Helgoland, the trends are significantly different from zero at
the 95 % confidence level. The mean increase of the annual maximum water level since 1950 is

about 20–40 cm at the coastal gauges and about 60–80 cm at the estuarine gauges Hamburg and
Bremen. The reasons for the increasement are not fully explored, but it could be the result of several
factors, such as mean sea level rise, variability in the wind climate, astronomical tide cycles and
hydro engineering measures.

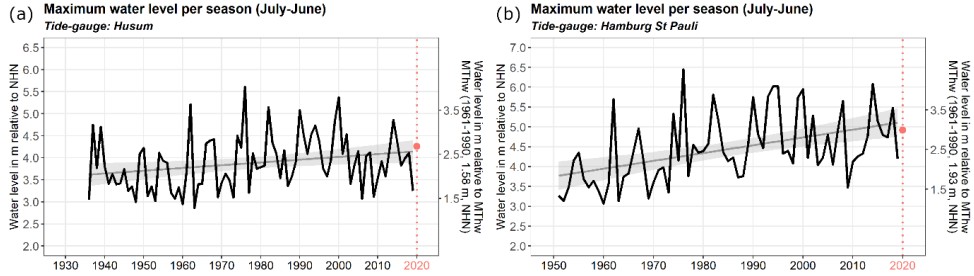


**Figure 6.** Maximum water level per season same as Fig. 3a but for the gauges (a) Husum and (b)
Hamburg St. Pauli.

Although the trends are positive, the annual maximum water level varies from season to season and

from gauge to gauge. In general, the annual maximum water levels are lower at Cuxhaven,
Helgoland, and Norderney, while they are higher at Husum, Hamburg, and Bremen. Among the
gauges, either the storm surge of February 1962 (Helgoland, Bremen, and Norderney) or the storm
surge of January 1976 (Husum, Cuxhaven, and Hamburg) is the highest since the beginning of data
availability. The highest water level since the 1950s occurred on 3 January 1976 at Hamburg St. Pauli

with about 4.5 m above MThw. In the last decade, the storm surge of December 2013 represents the
highest event. At some gauges (Cuxhaven, Hamburg, Bremen, and Norderney), it is among the five
highest events over the available periods.

*Frequency*



Annual storm surge frequency increased at all gauges (e.g. Fig. 4a and 7). Except for Helgoland, the
trends are significantly different from zero at the 95 % confidence level. In addition to the positive
trends, the number of events also varies from season to season and from gauge to gauge. In general,
there are fewer events at Helgoland, Cuxhaven, and Norderney, while events occur more frequently
at Husum, Hamburg, and Bremen. In the 1950s, about 1–3 storm surges usually occurred in a

season. Over the past few decades, the annual number of storm surges has increased by about one
at Helgoland and Norderney, and it has nearly doubled at Husum and Cuxhaven. At the estuarine
gauges, storm surge frequency has increased even more strongly. On average, there are about five
times as many storm surges per season nowadays than there were in the 1950s.

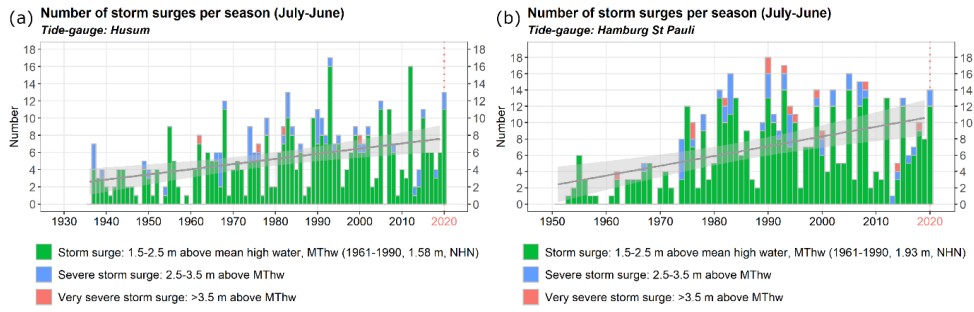


**Figure 7.** Number of storm surges per season same as Fig. 4a but for the gauges (a) Husum and (b)
Hamburg St. Pauli.

*Duration and intensity*

Due to the limited availability of high temporal resolution data, the statistics for storm surge
duration and intensity can only be evaluated for Cuxhaven. As introduced in Sect. 2.6, the total
duration of all events in a season and the intensity of the most intense event in a season are
evaluated (Figs. 5a and 5b). For both measures, upward trends significantly different from zero could
be inferred. Specifically, both measures have doubled since the 1920s. The annual total duration

increased from about 5 h to 12 h, and the annual maximum intensity increased from 1 m·h to about
3 m·h. Averaged over the reference period, the mean values of the total duration and the maximum
intensity were about 10 h and 2 m·h, respectively.

### 3.1.2 Assessment of the two seasons 2018/19 and 2019/20

*Height*
In season 2018/19, the annual maximum water levels at all gauges fell into the lowest category of
severity (1.5–2.5 m above MThw) (Fig. 6). This immediately implies that all events observed in the
season were minor (green marks in Figs. 8a and 8b). Except for Husum, the highest event occurred
on 8 January 2019, and its return period varies between 1 and 3 years depending on the tide gauge
(e.g. about 1 year for Hamburg in Fig. 8b). For Helgoland and Norderney, this was the only event
observed in the season. More events occurred at other gauges, but they only slightly exceeded the
thresholds (e.g. Husum and Hamburg in Figs. 8a and 8b). On average, such minor events occur about
twice a year, which are normal for these gauges in terms of storm surge height.

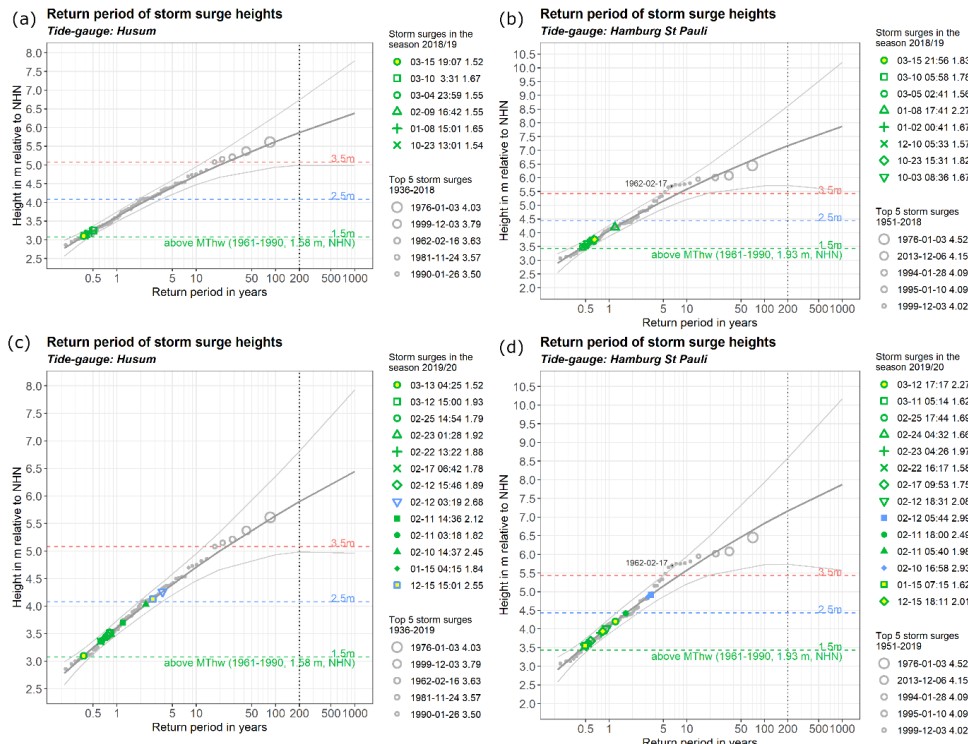

**Figure 8.** Return period of storm surges same as Fig. 3b but for the gauges (a, c) Husum and (b, d)
Hamburg St. Pauli (top: season 2018/19; bottom: season 2019/20).

Compared to season 2018/19, higher annual maximum water levels were observed in the following
season 2019/20 (Fig. 6). During 10–12 February 2020, the storm Sabine (Haeseler et al., 2020)
induced a series of consecutive storm surges. The highest water levels of the season were observed
during these days. The estimated return period of the highest event varies from 3 to 8 years at
different gauges (e.g. about 3.5 years for Husum and Hamburg in Figs. 8c and 8d). As the maximum



water level has increased over the past decades, the highest water levels of the season are

consistent with this long-term development. At Helgoland and Husum, the maximum storm surge height may serve as an example of this development since it corresponds to the average of the past years (Fig. 6a). In addition to this series of events, more minor events were observed at Husum, Cuxhaven, Hamburg, and Bremen. They are classified as minor events with estimated return periods shorter than one year, indicating that such minor events are common for these gauges.


*Frequency*

In terms of storm surge frequency, these two seasons 2018/19 and 2019/20 are significantly different from each other. In season 2018/19, storm surge frequency is around the average frequency of the reference period. A total of one or two events was observed at Helgoland,

Cuxhaven, and Norderney (Fig. 4a), whereas six or eight events were observed at Husum, Hamburg, and Bremen (Fig. 7).

In season 2019/20, the number of events was at least doubled at five out of the six gauges. Especially, the number of 13 events at Husum is remarkable and ranks among the top three highest

storm surge frequencies at this gauge (Fig. 7a). As the storm surge frequency has increased over the past decades, the higher storm surge frequency of this season is consistent with the long-term development. However, even compared to the mean annual frequency of the recent years, the number of storm surges in this season was much higher.

As the reference shows, the first storm surge ususally occurs in November or December, and most storm surges occur between November and February (Fig. 9). The course of season 2018/19 broadly followed the long-term median (Figs. 9a and 9b). For some gauges where frequencies eventually exceeded the median, the values are still below the corresponding 97th percentiles. Moreover, the numbers of events in the individual months were not exceptional.

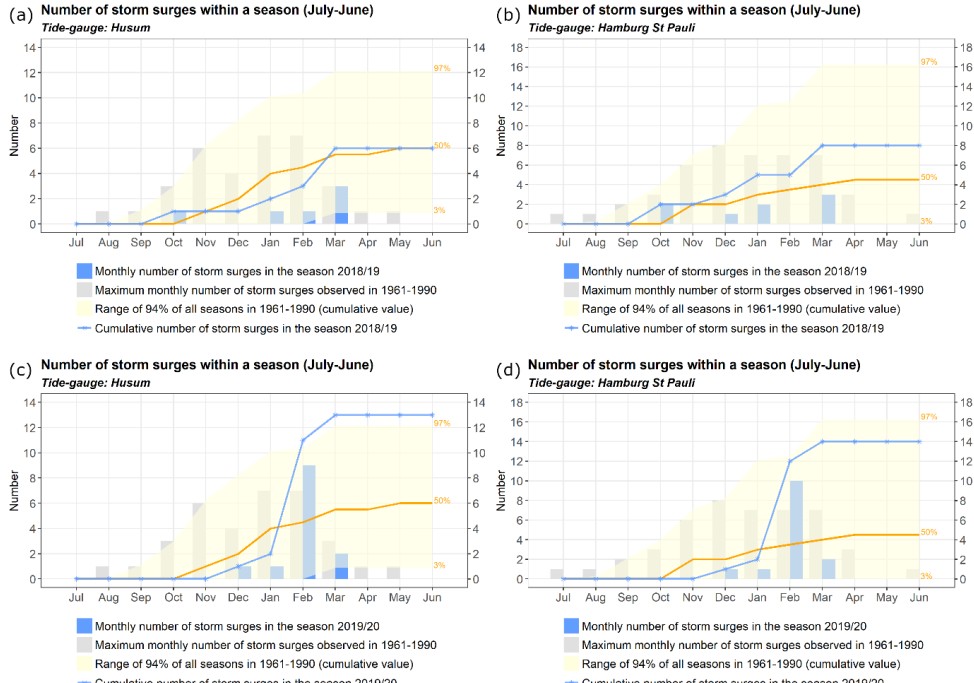

**Figure 9.** Number of storm surges within a season (July–June) same as Fig. 4b but for the gauges (a, c) Husum and (b, d) Hamburg St. Pauli (top: season 2018/19; bottom: season 2019/20).

In contrast, season 2019/20 was substantially different. The season started late initially, and the storm surge frequency was moderate and mostly below the long-term median. This character changed in February when the storm Sabine caused a record number of events in a relatively short period (Figs. 9c and 9d). For example, nine events (five caused by Sabine) were observed in February 2020 at Husum, which exceeded the maximum of seven events over the reference period (Fig. 9a).

Consequently, the cumulated number of events in the season exeeds the normal range; thus, it is an unusual season for Husum. Interestingly, this series of events occurred shortly after the conclusion of the project EXTREMENESS, which considered such a series as an unlikely event with potentially large consequences (Schaper et al., 2020; Weisse et al., 2020).

*Duration and intensity*

The two consecutive seasons also differ in terms of the total duration and the maximum intensity (Fig. 5). In season 2018/19, both measures were below the averages of the reference period, while they were significantly higher in season 2019/20. Especially for the total duration, long storm surge hours (20 h) occurred in season 2019/20, which could be attributed to a few long lasting events





and/or a larger number of shorter events in a season. As discussed above, in the case of season
2019/20, this is due to the latter. Regarding the long-term increase of storm surge duration at
Cuxhaven, the total duration in season 2019/20 corresponds to this development; however, it is
much higher than the average of recent years (Fig. 5a). The maximum intensity of the events in the
season is 2.7 m·hour (Fig. 5b). This corresponds to the increased maximum intensity of the recent

years and is thus exemplary for the long-term development of storm surge intensity at Cuxhaven.

In season 2018/19, the two minor events in January and in March at Cuxhaven lasted for about three
hours and one hour, respectively. The duration and intensity of both events were around the
corresponding median (figures not shown here). In season 2019/20, the duration and intensity of the

events were also within the normal monthly range (Figs. 5c and 5d). Though more events were
observed, only a few in February and the one in March are slightly above the corresponding median.
The rest of the events are shorter or less intense than about 50 % of the events observed in the
corresponding month over the reference period. In general, the storm surges at Cuxhaven were not
exceptionally long or intense throughout both seasons.


In summary, for the North Sea gauges, season 2018/19 can be considered as a typical storm surge
season in all aspects. All the storm surges were minor events, and their return periods are mostly
shorter than one year. Although the number of events at some gauges is slightly above the average
level, it is still far from the upper bound of the normal range. Season 2019/20 was more active and

unusual in some aspects, i.e., the height, frequency, and total duration of the events. It was
characterized by a slow onset, which was more than compensated by an unusual series of events in
February. In consequence, the number of events is around the upper bound of the normal range,
and at some gauges it was exceptional from both monthly and seasonally perspectives. The total
duration of the storm surges in the season was also doubled compared to the reference period.

**3.2 Baltic Sea coast**

3.2.1 Long-term development of storm surges

*Height*

At the German Baltic Sea coast, the annual maximum water levels show strong variability over the
available period. Linear trends within this period vary in their signs from gauge to gauge, and none of

them are significantly different from zero at the 95 % confidence level (Fig. 10). This is consistent
with the results based on annual data, which cover a longer period (e.g. Meinke, 1999). Since sea
level has risen at the German Baltic sea coast (Weisse and Meinke, 2016) and is expected to rise


further in the future (Grinsted, 2015), it cannot be excluded that the annual maximum water levels
may increase in the future as well. Thus, an ongoing monitoring is important, although no significant
trend in annual maximum water level has been found so far.

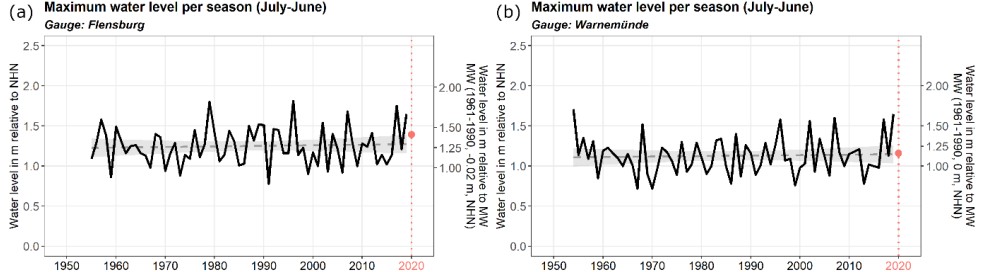

**Figure 10.** Maximum water level per season and its linear trend at the gauges (a) Flensburg and (b)
Warnemünde. The trend (gray line) is shown with the 95 % confidence interval (light gray band) and
as a doted line when it is not significant at the level of 95 %. The red dot denotes the maximum
water level observed in the current season (season 2019/20 here).

Over the past seven decades, the annual maximum water level varies between 0.7 m and 2.0 m
above MW at the analyzed gauges. The average maximum water level is around 1.2 m above MW.
The highest storm surge at the German Baltic Sea coast since the beginning of the measurements
took place on 13 November 1872 (Jensen and Müller-Navarra, 2008; Rosenhagen and Bork, 2009;
von Storch et al., 2015; Weisse and Meinke, 2016). The maximum water level was about 3.3 m above
MW on the southwestern coast of the Baltic Sea, which is much higher than any extreme event
occurred later on in this region. It was a very rare event, which, however, could occur any time
again. Since the 1950s, none of the water levels at the analyzed Baltic Sea gauges exceeded 2 m
above MW (Fig. 10). However, 2 m above MW was almost reached on 4 January 1954 at
Travemünde with a maximum water level of 1.97 m above MW and on 4 November 1995 at Kiel with
a maximum water level of 1.96 m above MW. The latter event is among the five highest events at all
analyzed Baltic Sea gauges since the 1950s, and it has been the highest event of Flensburg and Kiel
since then. At Warnemünde and Travemünde, the highest event in that period has occurred on 4
January 1954.

*Frequency*

The storm surge frequency at the analyzed Baltic Sea gauges shows pronounced interannual and
decadal variability. Over the available periods since the 1950s, the frequency varies between zero



and five events per season at Warnemünde, and between zero and nine events per season at other

gauges (e.g. Flensburg and Warnemünde in Fig. 11). On average, there were normally about one

event per season at Warnemünde and about two to three at Flensburg, Kiel and Travemünde. At

Travemünde and Warnemünde, the storm surge frequency shows a slight but not significant

increase over the available periods. This result is consistent with other studies, which analyzed other

periods, e.g. 1883–1997 (Meinke, 1999) and 1948–2011 (Weidemann et al., 2014). However,

contrary to the Monitor, the impact of sea level rise has been excluded in those studies. At all

analyzed gauges, the number of medium storm surges in the last three decades (1991–2020) is

lower than in the reference period (1961–1990). In the period of 1991–2020, five severe storm

surges occured (in 1995, 2002, 2006, 2017 and 2019). Compared to the reference period, the

number of severe storm surges has decreased at Kiel. However, it has not changed at Flensburg (Fig.

11a) and Travemünde and notably increased at Warnemünde (Fig. 11b).

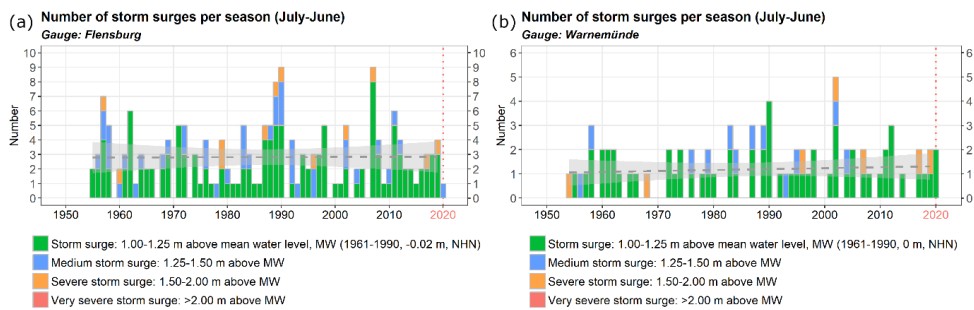

**Figure 11.** Number of storm surges per season and its linear trend at the gauges (a) Flensburg and

(b) Warnemünde. The color of the bars denotes the degree of severity (green – minor; blue –

medium; orange – severe; red – very severe). The trend (gray line) is shown with the 95 %

confidence interval (light gray band) and as a doted line when it is not significant at the level of 95 %.

The current season (season 2019/20 here) is highlighted with the red-dotted line.


*Duration and intensity*

Information on duration and intensity can be derived at all selected Baltic Sea gauges, since hourly

data are available. The total duatarion and the maximum intensity of storm surges per season show

strong variabilities and slightly negative trends, which are not statisticaly different from zero on a

95% confidence level (Fig. 12). In other studies, the absolute or maximum duration of single storm

surges above a certain threshold or within a certain range of different thresholds were analyzed (e.g.


Weidemann et al., 2014). These studies show a slight but not significant increase in storm surge
duration.

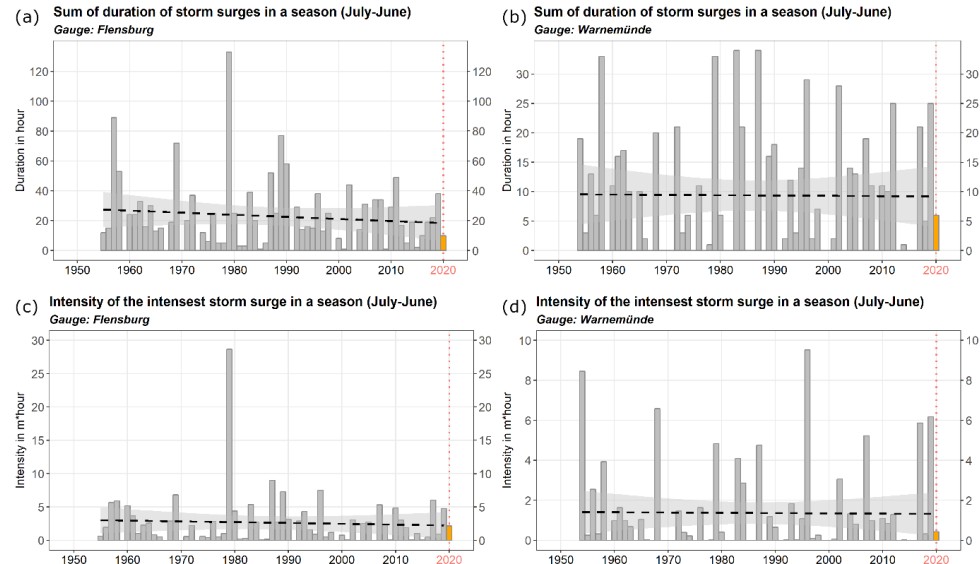


**Figure 12.** Total duration of storm surges per season (top) and the intensity of the intensest storm
surge per season (bottom) at the gauges (a, c) Flensburg and (b, d) Warnemünde. The gray bars
represent the past seasons, while the orange bar denotes the current season (season 2019/20 here).
The trend (black line) is shown with the 95 % confidence interval and as a doted line when it is not

significant at the level of 95 %.

The average of the total duration per season is about 20–30 h at Flensburg, Kiel and Travemünde
and about 10 h at Warnemünde. The highest total duration per season at Flensburg, Kiel and
Travemünde occured in season 1978/79, reaching 120–130 h. At Warnemünde, there was no such

season with extraordinary long duration of storm surges. Instead, the highest total duration per
season lasted 30–35 h, and all these seasons occurred before 1990. The average intensity of the
most intense event is about 2–4 m·h. At Flensburg, Kiel and Travemünde, the most intense event
during the analyzed period occurred in season 1978/79 with about 20–30 m·h. At Warnemünde, it
occurred in season 1995/96 with the intensity of about 10 m·h.


### 3.2.2 Assessment of the two seasons 2018/19 and 2019/20

*Height*


Assessing the maximum water levels of the two seasons, it is obvious that the maximum water levels

of season 2018/19 are higher than normal at all analyzed Baltic Sea gauges, whereas the maximum

water levels of season 2019/20 are close to the averages (e.g. Flensburg and Warnemünde in Fig.

10). Specifically, as shown in Figs. 13a and 13b, the highest storm surge of season 2018/19 occurred

on 2 January 2019, which was induced by the storm Zeetje (Perlet, 2019). The maximum water levels

reached 1.67 m, 1.65 m, 1.70 m, and 1.65 m above MW at Flensburg, Kiel, Travemünde, and

Warnemünde, respectively. For all gauges, these water levels are classified as severe storm surges

(orange mark) in the return period plot (e.g. Flensburg and Warnemünde in Figs. 13a and 13b). The

estimated return period of this severe event is 50–60 years at Warnemünde (Fig. 13b) and 10–20

years at the other three gauges (e.g. Flensburg in Fig. 13a). In addition to this severe event, there

were other minor events in this season (green marks). The estimated return periods of these minor

events are about 0.5–2 years, which are normal at these gauges.


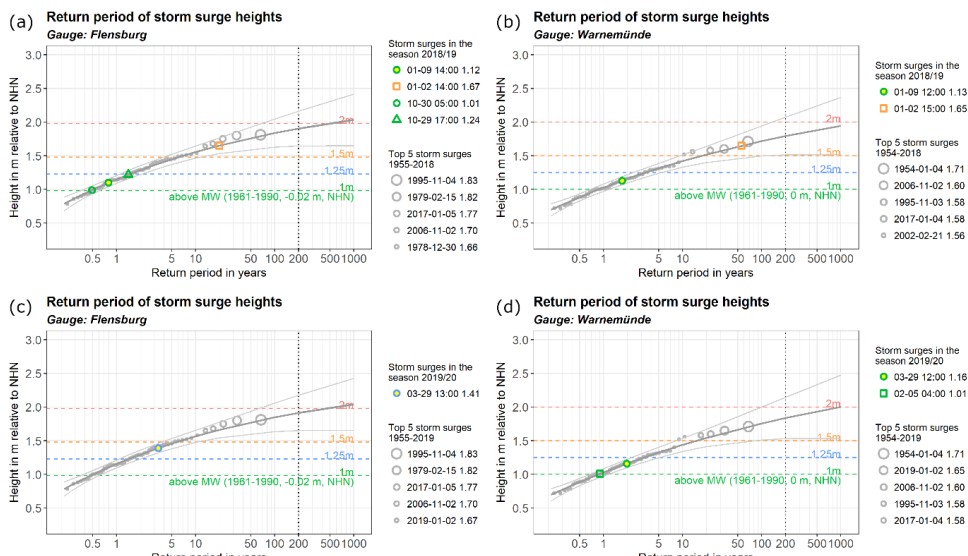

**Figure 13.** Return period of storm surges at the gauges (a, c) Flensburg and (b, d) Warnemünde. The

events of the "current" season (top: season 2018/19; bottom: season 2019/20) are marked by

colored symbols according to their heights and the respective four severity classifications (green –

minor; blue – medium; orange – severe; red – very severe). The estimated distribution (dark gray

curve) of the annual maxima (gray points) over the previous seasons was derived using the

generalized extreme value (GEV) distribution and the maximum-likelihood estimation (MLE). The

gray band shows the 95 % confidence interval of the estimation. The top five severe historical events

in the available period are represented by gray open circles with size indicating their magnitudes.






In season 2019/20, the maximum water levels of most gauges fall into the second category of severity (1.25–1.50 m above MW) (e.g. Flensburg Fig. 13c), while the maximum water level at Warnemünde is in the lowest category (1.00–1.25 m above MW) (Fig. 13d). This highest event of the season occurred on 29 March 2020 at all gauges. It is classified as a medium storm surge (blue

marks) at Flensburg, Kiel, and Travemünde with an estimated return period of 3–4 years. At Warnemünde, it is denoted as a minor storm surge (green mark) with an estimated return period of 2 years. Note that the highest storm surge of the previous season on 2 January 2019 is now on the historical lists of the five highest events at Flensburg and Warnemünde, as the historical lists have been updated automatically when season 2019/20 began (Figs. 13c and 13d).


*Frequency*

In season 2018/19, the storm surge frequency at all analyzed gauges are above the average of the reference period but within the normal range. Taking Flensburg as an example, there were three minor events (green marks) and one severe event (orange mark in Fig. 11) among the four events in

the season. With these four events, the storm surge frequency is above the average frequency but still in the normal range (Fig. 14a, blue line above orange line but within the yellow area).

In season 2019/20, only one event occurred at Flensburg and Kiel, which lables the season less active in comparison to the median over the reference period (e.g. Flensburg in Fig. 14c, blue line below

orange line). At Travemünde and Warnemünde, two events were observed, which is equal to the median for Travemünde and above the median of one event for Warnemünde (e.g. Warnemünde in Fig. 14d, blue line above the orange line). This reveals that the same storm surge season can be less active (Flensburg and Kiel), normal (Travemünde) or more active (Warnemünde) at the German Baltic Sea coast, depending on the location of the respective gauge.




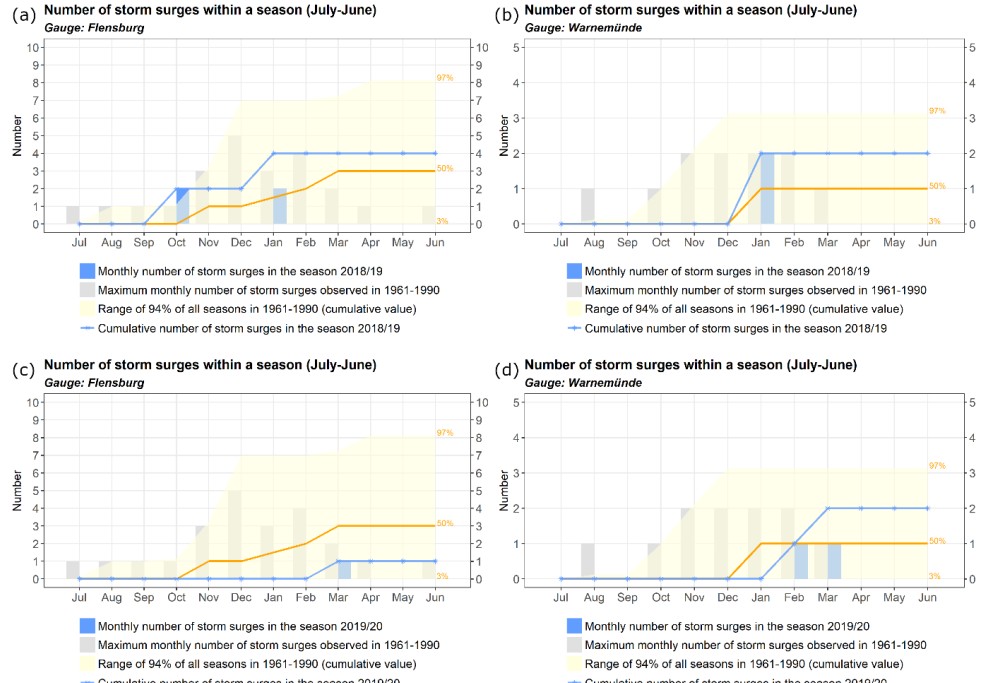

**Figure 14.** Number of storm surges within a season (July–June) at the gauges (a, c) Flensburg and (b, d) Warnemünde. Bars: the monthly number in the "current" season (top: season 2018/19; bottom: season 2019/20) is shown as blue bars, and the historical monthly maximum in the reference period 1961–1990 is shown as gray bars. Curves: the cumulative number of events in the current season is shown as a blue curve, in comparison with the 50th percentile (orange curve) and the range of the 3rd and 97th percentiles (yellow shaded are) over the reference period.

As the orange line in Fig. 14 indicates, the first storm surge of a season normaly occurs in November at Flensburg and in January at Warnemünde. The two events in October 2018 at Flensburg indicate a relatively early start of season 2018/19, whereas it was a rather late start in the following season, when the first and only event at Flensburg occurred in March (Figs. 14a and 14c). At Warnemünde, however, it was rather normal for both seasons, with the first event in January and February (Figs. 14b and 14d).

*Duration and intensity*

In season 2018/19, the total duration of storm surges is almost double the average over the reference period (e.g. Flensburg and Warnemünde in Figs. 12a and 12b). As discussed in Sect. 3.1.2, long duration could be caused by a few long lasting events and/or many shorter events. In this





season, both factors contribute to the long duration. As described above, the storm surge frequency
of this season is higher than normal. Moreover, several events lasted longer than the monthly
average of the reference period (Figs. 15a and 15b, red dot above the median). The storm surges in
October and January also show higher intensity than the monthly average of the reference period
(Figs. 15c and 15d).




**Figure 15.** Box plot of the duration (top) and the intensity (bottom) of storm surges in each month
for season 2018/19 at the gauges (a, c) Flensburg and (b, d) Warnemünde. Blue shows the statistics
(the 3rd, 50th, 97th percentiles, and the maximum) of the events in each month of the reference
period 1961–1990, while the red dots signify the events of the current season.

In particular, duration and intensity of the severe event on 2 January 2019 are between the
corresponding median and the 97th percentile of the January events over the reference period at all
gauges, except for the intensity at Warnemünde. As shown in Figs. 15b and 15d, although the
duration of the event is not exceptionally long, the intensity was as high as the 97th percentile, as
intensity reveals combined information of height and duration. For the event on 29 October 2018,
which mainly affected the western gauges, the duration and intensity exceed the maximum values of
all October events at Flensburg in the reference period (Figs. 15a and 15c).


In season 2019/20, the total duration and the maximum intensity of storm surges are mostly below

the long-term average (Fig.12), and the duration and intensity of the events are low in general (Fig.

16). At Flensburg, the duration and intensity of the event in March 2020 are higher than the medians

of the March events in the reference period, while the values are higher than the maxima at

Warnemünde (Fig. 16). Note that storm surges in March are common for Flensburg, but much rare

for Warnemünde, indicating the local differences among gauges.

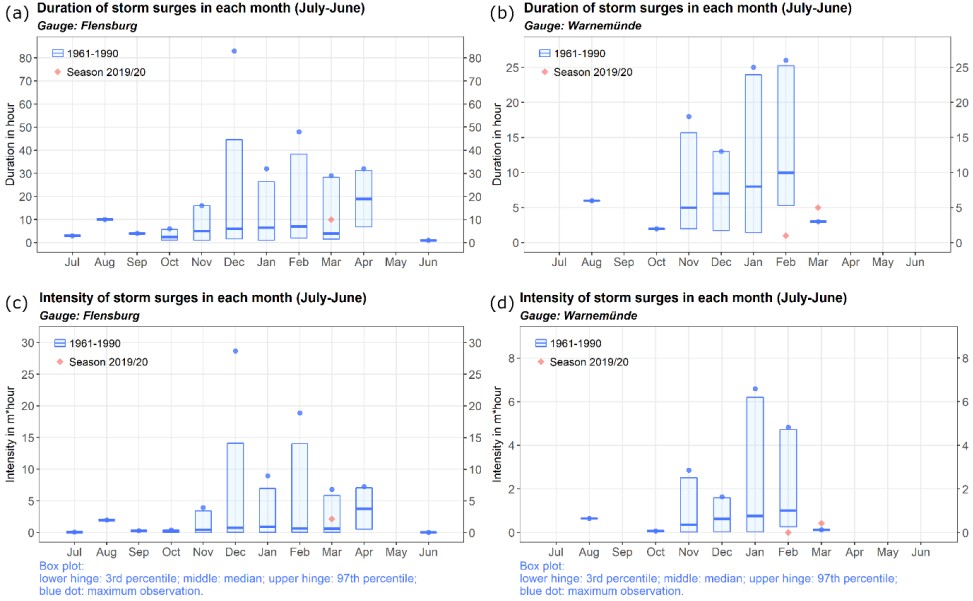

**Figure 16.** Same as Fig. 15 but for season 2019/20.

In summary, for the Baltic Sea gauges, season 2018/19 could be described as an active season in all

aspects, though it is still within the normal ranges derived over the reference period. This is

particularly due to the severe event on 2 January 2019, which is among the severest historical events

since the 1950s. In contrast, season 2019/20 has fewer storm surges with lower heights and shorter

duration, which is an example of a normal storm surge season for the Baltic Sea coast.

**4 Discussion and summary**

To monitor potentially increasing storm surge hazards and to assess changes in near real-time, we

propose an approach that puts recent extremes and the ongoing storm surge season into a historical

prespective. This approach was implemented into a prototype web tool (we called the Storm Surge





Monitor) for the German North Sea and Baltic Sea coasts. With the help of this Monitor, storm

surges at tide gauges are instantaneously detected and set into the long-term context of varying
storm surge activity. This way, not only an assessment of the current season and ongoing events is
achieved but also the development over the past seasons is documented. Understandable
information is provided to users to assess if and to what extent an event or a season is unusual
compared to the statistics of the extreme events in the past decades. Moreover, measures to assess

long-term changes can be inferred from the Monitor. Note that no attribution is made so far, but
only the extent to which the measures are within previously observed ranges is assessed.

The observed long-term changes at the German North Sea coast (e.g. the annual maximum water
level in Fig. 3a) could in principle originate from various factors, such as changes in storm activity,

astronomical tide cycles, or sea level rise. Locally, water works may also play a role. To date, there is
no clear evidence suggesting a significant long-term change of storm activity in this region (Feser et
al., 2015; Krieger et al., 2020; Krueger et al., 2019; Stendel et al., 2016; Weisse et al., 2012). It is
likely that the observed long-term changes are largely related to the local mean sea level rise (e.g.
Weisse et al., 2012; Woodworth et al., 2011). Moreover, it should be noted that the rising relative

sea level is not necessarily fully related to climate change, but partly due to non-climate factors, e.g.
land subsidence. As tide gauge is land-based, the observed water level contains the contribution
from crustal motions, which may be induced by natural phenomena (e.g. GIA) and local
anthropogenic activities (e.g. groundwater extraction) (Rovere et al., 2016; Stammer et al., 2013;
Tamisiea and Mitrovica, 2011).


At the analyzed gauges at the German Baltic Sea coast, no trend significantly different from zero at
the 95% confidence level has been found so far. As for the annual maximum water levels, this is in
accordance with the results of Meinke (1999). Slight but not significant trends have been found in
storm surge frequency at Travemünde and Warnemünde, in agreement with previous studies

(Meinke, 1999; Weidemann et al., 2014). The total duration and the maximum storm surge intensity
show slight decreases, which are not significantly different from zero at the 95% confidence level.
Beside these findings related to long-term trends, further monitoring is necessary. Although severe
storm surges are rare events and their number has mostly not changed, they have mayor impact to
coastal communities. In case of damages, the Monitor helps to distinguisch different sources: if

damages occur while the storm surge is within the normal range, it may indicate deficiencies of
actual coastal protection; if damages occur because the storm surge was unusually high or intense,
this indicates that coastal regions are not adequately prepared, even though the occurance of very


unusal events is plausible as the storm surge of November 1872 shows. Moreover, it is plausible that height, frequency, duration and intensity may change in the future, as sea level continues to rise at

the German Baltic coast (Grinsted, 2015; Weisse and Meinke, 2016). Thus, continuous monitoring on seasonal anomalies contextualized in the long-term development of storm surge characteristics is very important. After each storm surge season, the long-term trends are automtacally recalculated and tested if they are significantly different from zero at the 95% confidence level. This helps to contextualize local long-term statistics with future sea level scenarios and thus may serve as a

scientific basis for regional adaptation strategies of coastal ptrotection measures.

Discussing these two recent and subsequent seasons, it can be inferred that severity and activity vary across regions. A season can be normal or less active to one coastal region but more active or even unusual to another. This is due to the different regional meteorological conditions that are

responsible for generating the storm surges. For example, season 2018/19 was largely typical for the North Sea coast, while it was relatively active for the Baltic Sea coast mainly because of one severe event. Vice versa the following season 2019/20 was more active at the North Sea coast mostly because of a series of events in February, which did not cause corresponding events in the Baltic Sea due to the wind direction and the track of the storm systems. Furthermore, water levels at different

tide gauges also react differently, which depends largely on the local wind conditions and the shape of the coastline (Jensen and Müller-Navarra, 2008).

Nowadays, web-based applications are used to link scientific research and public demands. For sea level extremes, various efforts exist. Such tools provide online access to the statistics of extreme

water levels and document the severity and consequences of historical flooding. Examples are the Extreme Water Levels site from NOAA (https://tidesandcurrents.noaa.gov/est/), or the SurgeWatch site for the UK coast (https://www.surgewatch.org/) (Haigh et al., 2015). Building upon and adding to previous applications, our tool focuses on the near real-time contextualization of the current season and its events against the background of long-term variability and change with the intention

to provide an up-to-date and continuous piece of coastal climate services. Looking back in time helps to improve the understanding of the past and to better evaluate the state of the present. Monitoring the present enables us to update the statistics and to detect the changes at the earliest possible stage.

The Monitor and the statistics are freely available online. They are expected to be useful and meaningful to the public and in particular to the media who are concerned about storm surges. It is



also relevant to the multi-sector coastal stakeholders who demand this information for coastal flood risk management and planning. Moreover, it could serve for educational purposes, for example, illustrating storm surge activity at the German coasts. Last but not least, it can be useful to

researchers as auxiliary information in the presentations of their scientific results, or as pre-knowledge for further research especially on the most recent extreme events.

The up-to-date information in collecting and analyzing storm surges is currently in need at local to regional scales for coastal climate service. This work has demonstrated a way to make real-time data

more meaningful and accessible to the public as well as a way to deliver more up-to-date information. It has the potential to be developed into a larger tool including a network of tide gauges that covers the coasts under threat of storm surges. Our attempts started with the storm surges on the German coasts and focused on event detection and description. We propose that the concept can also be used to other variables (e.g. sea level and storm activity), and can be developed further

to include attribution.

## Author contributions

RW and IM initiated the idea of the web tool and designed it. XL performed the analysis and programmed the web tool. All authors contributed to the preparation of the manuscript.

## Competing interests

The authors declare that they have no conflict of interest.

## Acknowledgment

The map in Fig. 1 is generated by Leaflet | © OpenStreetMap contributors. This work is a contribution to the project "European advances on CLImate Services for Coasts and SEAs" (ECLISEA) funded through the ERA4CS framework (European Research Area for Climate Services).

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
