# Peer review of "Still normal? Near-real-time evaluation of storm surge events in the context of climate change"

_Natural Hazards and Earth System Sciences, 2021_

## Author Response (AR1)

**Reply on RC1**

We appreciate the constructive comments of the reviewer. They helped us to clarify and improve our manuscript. In the following, we show how we addressed the points raised by the reviewer in the revised manuscript.

In the general discussion of the manuscript, the reviewer wondered whether a scientific journal would be the right place to present and discuss the developed service. We argue, that our manuscript describes and establishes a new method, which is well within the scope of a scientific journal. More specifically, we chose Natural Hazards and Earth System Sciences as journal, as its scope includes

- *"the detection, monitoring, and modelling of natural phenomena, and the **integration of measurements** and models **for the understanding** and forecasting **of the behaviour and the spatial and temporal evolution of hazardous natural events** as well as their consequences;"*

    and

- *"the design, development, experimentation, and validation of **new techniques**, methods, and **tools for the detection, mapping, monitoring, and modelling of natural hazards** and their human, environmental, and societal consequences;".*

From our perspective, the storm surge monitor and our manuscript are fully in-line with these points and the scope of the journal.

However, we agree with the reviewer that our points were not made clear enough to be easily taken up. We substantially re-wrote and partly re-structured the manuscript to make these points clearer and more explicit. We also followed the suggestion of the reviewer and shortened the suggested parts (originally section 3) and extended our efforts to explain the approach and to interpret the data.

The specific points of the reviewer were addressed as follows:

- Lines 94, 121: The typos were corrected.
- Line 223: We agree that the choice was somewhat arbitrary. We checked and found that no events occurred for which this de-clustering had an effect. We removed this part from the text and the analysis.
- Line 227: Was modified as suggested.
- Lines 279/280: We followed the suggestion and adopted captions throughout the manuscript and shifted general discussions and descriptions into the text.
- Line 289: The typo was corrected.
- Line 299: The text was changed accordingly.
- Lines 308, 313 (318), 328, 371, 384 (386): Was corrected as suggested.
- Lines 386-388: We agree with the reviewer. We added references and extended the discussion in the revised manuscript.
- Lines 486-488: The reviewer is right that heights were modest. But the results from the cited literature showed that particularly a series of events even of modest height may challenge coastal protection. We are more explicit about this in the revised manuscript.
- Lines 523-527: Mean sea level is rising in the Baltic Sea but does show a strong spatial gradient with smaller values along the German Baltic Sea coast (see e.g., https://doi.org/10.3389/fmars.2021.647607). Moreover, there is strong decadal variability in sea level trends and large interannual and decadal variability. We extended the discussion in the revised manuscript and put the statements into context.

- Lines 560-562: We took up this point raised by the reviewer and extend the discussion accordingly.
- Line 567: The text was revised accordingly.
- Lines 571-573: We carefully checked the manuscript for repetitions and removed them where not required within the context.
- Lines 580-583: As stated in the text, the increase described in cited literature was not significant. We introduced a corresponding discussion in the text.

**Reply on RC2**

We appreciate the constructive comments of the reviewer. They helped us to clarify and improve our manuscript. In the following, we show how addressed the points raised by the reviewer in the revised manuscript.

The reviewer makes the following specific comments (in italic) that we addressed as follows (in blue).

*"Depending on the temporal resolution of the data, not all graphs (duration and intensity) can be shown. This is mentioned in the manuscript but I couldn't find it in the Monitor.*

A corresponding statement was added to the monitor and goes online with the next revision.

*"Chapter 3 describes five stations … in detail. While this is in general interesting, it occurs a bit lengthy and redundant, especially where information/comparison to other stations is given but not shown. I wonder if some information could also be summarized in a table (e.g. trends) which would make the description a bit more compact. Instead, I would like to see some emphasis on different evolutions and signals!"*

We substantially shortened the description and partly re-structured it to be more easily accessible. We further focused on the main evolutions and signals following also a similar suggestion by reviewer #1. We also summarize the results as suggested here by reviewer #2.

*"On page 16, line 407-409 the authors discuss the occurrents of events at stations Helgoland, Cuxhaven, Norderney (fewer) and more frequently at Husum, Hamburg and Bremen. I wonder, why Cuxhaven has fewer events while located at the estuarian tip between Weser and Elbe?"*

There are primarily two reasons. First, the specific configuration of the coastline and bathymetry makes a tide-gauge more/less susceptible to storm surges. Also, the wind direction that most effectively generates storm surges differs between the tide gauges. Second, the Monitor uses a common threshold for all tide-gauges that is used by the Federal Maritime Agency (BSH) to issue storm surge warnings. In other statistics, local thresholds are used that are defined according to the DIN 4049-3. This will lead to differences in the number of detected surges. The issue is taken up and described in the FAQ section of the monitor. To address this comment, we also discuss this more clearly and explicitly in the revised manuscript. Taking feedbacks from stakeholders into account, we plan to include both statistics in the next version of the Monitor. This is now also discussed in the manuscript.

*"I was wondering, if the authors have assessed the demand and need for such a Monitor and if they got into contact with key stakeholders to discuss the usefulness, the design, and demand for the Monitor and the provided information? In times of modern knowledge exchange, we do know how important co-design of such processes is and the early involvement of potential users and stakeholders. A short additional paragraph on this aspect would be worthy for the readers and other scientists who plan similar services."*

The Monitor was developed in close contact with representatives of authorities responsible for coastal protection. We constantly receive feedback (see the reply on the comment above) and plan to include such feedback in new releases. Presently a survey is performed among stakeholders in the region the feedback of which will be evaluated and

taken up to improve the Monitor. As suggested by the reviewer some additional text was included in the last section to briefly discuss these issues.

*"The title of the manuscript is really long? Is this really needed or could it be shortened to make it handier?"*

We agree with the reviewer and replaced the title with a shorter one. The title now reads: Still normal? Near-real-time evaluation of storm surge events in the context of climate change

The reviewer further suggested several technical corrections that were addressed as follows:

- Lines 94, 121: The typos were corrected.
- Figure 2, Line 223: We agree that the choice was somewhat arbitrary. We checked and found that no events occurred for which this de-clustering had an effect. We removed this part from the text and the analysis.
- Table 2: Explanation was added.
- Figure 3b: Explanation was added.
- Lines 420-427, 490-500, 591-599: We unified as suggested.
- Lines 460-461: Was modified as suggested.
- Line 739: The typo was corrected.
- Figure captions were revised taking also the comments from reviewer #1 into account.
- References were checked, corrected, and updated.

---

## Author Response (AR2)

**Reply on RC1**

The authors did a good job in implementing my suggestions on the first version of their paper. The paper is now more concise and better suited for a scientific journal.
While browsing through the paper I potted a few typos:
line 481: Figure 9c should read Figure 12c
line 523: sown -> shown
line 554: 2918 -> 2018
line 589: delete ' the' at the end of the line

REPLY: All corrections have been made as suggested.

**Reply on RC2**

General comments:
The authors provided a thoroughly revised version of their original manuscript, addressing my given comments of the earlier version. They put major efforts in reformulations of the text, shortened and re-structured the manuscript which makes it much more focused and easier to follow, now. They now stick more to the general aspects and evolution of the monitor and its statistics.
I therefore suggest publication of this manuscript after correction of some minor technical aspects which I list in the following:
Line 8: check formulation of the sentence: …, in which the extent to which the event was ….
REPLY: The sentence has been revised.

Line 39: Weisse et al. 2009 ◊ Reference missing
REPLY: The reference has been corrected.

Line 81: Kodeih 2019 ◊ should be Kodeih et al 2019?? Check!!!
REPLY: The references were corrected.

Line 193: MTHws and MWs are not introduced, avoid unnecessary abbreviations
REPLY: Abbreviations were removed.

Line 391: Kalén 2021 is not referenced but Kalén 2020 .. check
Same on line 595
REPLY: The references were corrected.

Line 420: Weidemann et al 2014 should be Weidemann 2014
Same on line 592
REPLY: The references were corrected.

Line 705: check if the year is correct, in the paper it is cited as of 2021
REPLY: The reference was corrected.

Line 730: still not resolved

Kodeih, S., et al.,: Climate information needs from multi-sector stakeholders, Deliverable 1.B, Work Package 1, Project ECLISEA, http://www.ecliseaproject.eu/wpcontent/uploads/2019/2008/d2011.b_report_stakeholder-needs_final_20180629.pdf, 2018.

⌐ Check weblink: page couldn't be found and add all co-authors

REPLY: Corrections have been made.